# Enteral Resuscitation: A Field-Expedient Treatment Strategy for Burn Shock during Wartime and in Other Austere Settings

Ian F. Jones [1,*], Kiran Nakarmi [2], Hannah B. Wild [3], Kwesi Nsaful [4], Kajal Mehta [3], Raslina Shrestha [2], Daniel Roubik [5] and Barclay T. Stewart [3,6]

1  Madigan Army Medical Center, Tacoma, WA 98431, USA
2  Nepal Cleft and Burn Center, Kirtipur 44600, Nepal; raslinas@uw.edu (R.S.)
3  Department of Surgery, University of Washington, Seattle, WA 98195, USA; kajalm@uw.edu (K.M.)
4  Department of Plastic, Reconstructive Surgery and Burns Unit, 37 Military Hospital, Accra GA008, Ghana; knsaful@yahoo.co.uk
5  United States Army Medical Corps, San Antonio, TX 98234, USA
6  Harborview Injury Prevention and Research Center, Seattle, WA 98104, USA
*  Correspondence: ian.f.jones.mil@health.mil

**Abstract:** Burn injuries are a constant threat in war. Aspects of the modern battlefield increase the risk of burn injuries and pose challenges for early treatment. The initial resuscitation of a severely burn-injured patient often exceeds the resources available in front-line medical facilities. This stems mostly from the weight and volume of the intravenous fluids required. One promising solution to this problem is enteral resuscitation with an oral rehydration solution. In addition to being logistically easier to manage, enteral resuscitation may be able to mitigate secondary injuries to the gut related to burn shock and systemic immunoinflammatory activation. This has been previously studied in burn patients, primarily using electrolyte solutions, with promising results. Modern ORS containing sodium, potassium, and glucose in ratios that maximize gut absorption may provide additional benefits as a resuscitation strategy, both in terms of plasma volume expansion and protection of the barrier and immune functions of the gut mucosa. While enteral resuscitation is promising and should be used when other options are not available, further research is needed to refine an optimal implementation strategy.

**Keywords:** burns; resuscitation; gut physiology; enteral resuscitation; oral rehydration solution; low- and middle-income countries

## 1. Introduction

Major burn injuries that occur during war and disaster represent a significant challenge for low-resource and/or disrupted health systems. The resources needed to adequately resuscitate and care for burn-injured patients can quickly overwhelm those available in austere settings. In addition to a lack of burn care expertise, the large cube (i.e., weight, volume, size) of sterile intravenous (IV) fluids required for burn-injured patients in field units and far-forward settings remains a major limiting factor of austere burn care. Military medicine has sought innovative technical solutions to this in the form of various man-portable IV fluid makers and alternate resuscitation fluids, including colloids (e.g., starches, lyophilized plasma) or hypertonic saline, with only partial success. One strategy for alleviating IV fluid requirements is enteral resuscitation (EResus) with oral rehydration solution (ORS). The use of ORS in the treatment of another disease characterized by massive fluid losses, cholera, is already considered one of the greatest public health advances in modern times. While the evidence base is small, it has also shown success when applied to the resuscitation of burn-injured patients.

## 2. Battlefield Burn History and Epidemiology

Burn injuries are a ubiquitous threat on the battlefield, occurring in 5–20% of all casualties in conventional warfare [1]. The incidence of military burn injuries varies based on the setting and types of units involved in a conflict. Mechanized warfare between armored units and naval warfare in particular result in high rates of burn injuries. This is evident when comparing the rate of burn injuries from the Vietnam War (4.6%), fought largely by infantry, to the heavily mechanized 1973 Arab–Israeli War (10.5%) and the 1982 Lebanon War (8.6%) [2,3]. In the Falklands War, 34% of British naval casualties suffered burn injuries [4].

Advances in military clothing and vehicle technology have resulted in fewer and less severe burns among military personnel. This is evidenced by the reduced rate of burn injuries in Iraq and Afghanistan after the introduction of fire-resistant clothing and mine-resistant, ambush-protected vehicles to US forces in 2007 [5–7]. However, a large conflict between mechanized forces is estimated to generate thousands of moderate-to-severe (20–90% total body surface area (TBSA)) burn casualties requiring resuscitation [8,9].

Moreover, advances in protective equipment do not mitigate civilian harm in conflict settings. Civilians living in war zones also suffer from burn injuries caused by mechanisms including the use of explosive weapons in populated areas, incendiary weapons, and explosive ordnance. Increasingly, civilians are targeted by hostilities and now comprise up to 80% of those killed during war [10–12]. In addition to direct attacks on civilians and injuries caused by explosive ordnance, injuries also result from the deterioration of infrastructure and public safety [13]. In Baghdad during the Global War on Terror, the majority of burn injuries were not directly conflict-related but rather due to the degradation of infrastructure (e.g., electrical wires, unsafe cooking devices) and breakdown in usual safe behaviors [14]. Damage to the healthcare system, which is increasingly intentional, further compounds the challenges that civilians face during conflict [15].

## 3. Logistical Constraints

One of the main differences between being wounded on the battlefield and during peacetime is the delay in both stabilization and definitive care. It frequently takes hours to days to reach a specialized medical unit during prolonged-field-care scenarios or shipboard incidents, and days to weeks to evacuate a casualty from the point of injury to a burn center [16]. During this period, casualties are cared for with extremely limited resources, often only with the items the provider is able to carry in an aid bag. The fluid requirements for even a moderate-size burn cannot be transported by person for use in these scenarios.

Further, it can be difficult to obtain the IV or durable intraosseous access necessary to safely administer large volumes of fluids. A retrospective review of prehospital interventions performed for casualties during Operation Iraqi Freedom found that 40% of casualties arrived at combat support hospitals without IV access [17]. Only 50% of casualties with major burn injuries (i.e., >20% TBSA) had prehospital IV access, and only 85% of those with IV access had resuscitation initiated prior to arrival [18]. This delay in resuscitation is alarming, particularly for casualties with large burns, since time to resuscitation is a major predictor of mortality and even long-term health-related quality of life [19].

EResus can mitigate some of these logistical challenges. First, EResus can be accomplished with sachets of oral rehydration solution reconstituted with locally sourced potable (not sterile) water. Therefore, large volumes of sterile crystalloid solutions are not required. A 10 lb package of ORS is about the size of five bags of IV fluid but yields 125 L of fluid. The equivalent amount of IV fluid would weigh 287.5 lbs. Second, EResus does not require IV access. Fluids can be administered by drinking with or without buddy support. Lastly, if a patient is incapacitated or unable to drink, EResus can be administered via a nasogastric tube placed without advanced equipment.

## 4. History and Physiology of Enteral Resuscitation

EResus relies on the efficient enteral absorption of fluids and dissolved solutes (e.g., sodium, potassium, glucose). Under normal physiologic conditions, the bioavailability of water and electrolytes approaches near 100%, with only minimal amounts excreted in stool and a large reserve of excess absorptive capacity [20]. This excess absorptive capacity can be leveraged for the treatment of various disease states that cause dehydration.

The classic example of the lifesaving treatment potential of EResus is seen during acute watery diarrhea (e.g., cholera). During the 1831 European cholera epidemic, O'Shaughnessy and Latta developed a method of treating the profound dehydration caused by cholera with intravenous fluids [21]. Intravenous fluid resuscitation remained the mainstay of treatment for cholera until the 1970s, when enteral resuscitation with ORS came to prominence in South Asia.

The science behind enteral resuscitation rests on two physiological phenomena: (i) the independence of the absorptive and secretory functions of the gastrointestinal tract, and (ii) the cotransport of sodium and glucose by the sodium–glucose cotransporter (SGLT1) [22]. The discovery of sodium–glucose cotransport has been touted as "the most important medical advance of [the 20th] century" [23]. These findings, which were discovered in the 1950s, were not initially known to the physicians who would eventually develop ORS in the 1960s [24]. Early studies on EResus for people with cholera by US Navy researchers serendipitously used glucose to maintain the osmolality of the resuscitation solution. It was quickly recognized that glucose dramatically increased the absorption of sodium and water as well. This resuscitation strategy was further refined at the SEATO-Pakistan Cholera Research Laboratory in Dhaka and the Johns Hopkins Center for Medical Research and Training in Calcutta. This culminated in the work by Mahalanabis treating refugees of the Bangladeshi War of Independence, which demonstrated the effective use of ORS to treat cholera patients in a severely resource-limited field setting, leading to the widespread adoption of ORS around the world [24,25]. Since this report in 1968, the mortality from diarrhea in children under the age of five has dropped from 4.6 million per year to 500,000 [26].

## 5. Gut Physiology in Burns

The massive inflammatory response produced following a major burn injury leads to a systemic insult with multiorgan effects. The vasculature is one of the most profoundly and noticeably affected, with increased permeability leading to widespread edema and loss of intravascular volume. This loss of circulating blood volume, combined with variable degrees of myocardial dysfunction and altered systemic vascular resistance, engenders burn shock [27–30].

Burn shock results in splanchnic vasoconstriction and a nearly 50% reduction in blood flow to the GI tract [31,32]. Autopsies of burn-injured people found that more than 50% had evidence of mucosal ischemia and necrosis, which is consistent with endoscopic findings in living burn-injured patients [33]. This has a wide range of negative effects on the function of the entire GI tract, including global decreases in motility from the stomach to the colon [34–39]. More concerning, however, is the potential for increased infectious complications and immunoinflammatory activation.

After a burn injury, there is a breakdown of the physical barrier functions of the intestinal mucosa. In experimental models, increased intestinal permeability and histologic changes can be detected as early as one hour after injury, with decreased tight-junction protein synthesis and more severe histologic changes (e.g., necrosis, epithelial loss) at two hours after injury [40]. This increase in intestinal permeability has also been seen in burn-injured patients and correlates with burn size [41,42]. There is also evidence of pathological shifts and general collapse of the intestinal microbiome with early proliferation of more pathogenic bacterial species (e.g., Gram-negative rods, yeasts), which leads to further mucosal barrier immune dysfunction [43–46].

Collectively, these mucosal and microbiological changes allow for the translocation of bacteria and endotoxins into lymphatic and ultimately the central circulation, where they are disseminated throughout the body and potentiate organ dysfunction (e.g., acute respiratory distress syndrome, endothelial injury, cardiovascular dysfunction). The translocation of bacteria across the small intestine has been observed in animal models of burn injury with evidence of migration to the mesenteric lymph nodes, liver, spleen, kidneys, and lungs [32,44,47]. Although this gut–lymph translocation pathway has not been examined in burn-injured patients, a study of surgical patients undergoing laparotomy demonstrated translocation of bacteria into the mesenteric lymph nodes [48]. It is hypothesized that this translocation of intestinal bacteria and endotoxins is a key driver of sepsis and multisystem organ dysfunction in burn-injured patients [49–51].

Early enteral nutrition plays a key role in mitigating the damage to the gastrointestinal tract after a burn injury. Experimental models suggest that the gut mucosa is primarily fueled by luminal nutrients, and early enteral nutrition can help preserve gut barrier function, resulting in decreased bacterial translocation [52,53]. A trial of early enteral nutrition in burn-injured patients suggested that burn-shock-induced reduction in intestinal blood flow is reversible with early enteral feeding [54]. This improvement in intestinal blood flow and the resultant preservation of mucosal barrier function could explain the reduction in mortality, gastrointestinal hemorrhage, sepsis, and pneumonia in burn-injured patients who receive early enteral nutrition [55]. While ORS lacks many of the components of enteral nutrition formulas, the nutritional support it does contain may provide some of the benefits seen with early enteral nutrition.

Another key difference between EResus and enteral nutrition is that the volumes needed for EResus are significantly higher than those for enteral nutrition (e.g., 300–1200 mL/h versus 50–100 mL/h). The success of EResus relies on the ability of the stressed and damaged gastrointestinal tract to adequately absorb this volume. Fortunately, animal models and extensive experience with the EResus of patients with acute watery diarrhea have demonstrated that the gastrointestinal tract can absorb up to 20 mL/min. The US Army Institute of Surgical Research demonstrated that the gastrointestinal tract of burned pigs was able to absorb volumes commensurate with the Parkland formula (4 mL/kg/% TBSA burned) and with an efficiency similar to IV crystalloid [16]. There is also some evidence that Eresus has positive immunomodulating effects in animal models, but it is unclear if the magnitude of the benefit will be similar to those already receiving early enteral nutrition [56].

## 6. History of Enteral Resuscitation for People with Burn Injuries

Fluid losses from burn injuries are less visible than those from cholera. This led to the toxin theory of burn shock, which prevailed well into the 20th century. It was not until the Rialto Theatre Fire in 1921 that the toxin theory was challenged by Underhill [57]. Drawing on his experience with chemical-weapon victims during World War I, he noticed the marked hemoconcentration of his burn-injured patients and concluded they were suffering from severe fluid losses from damaged tissues [57]. He concluded that, like other diseases characterized by massive fluid losses, including cholera, the treatment should be "the forcing of fluid by whatever channel possible", mainly intravenous but also via oral, subdermal, and rectal routes [57,58].

Another disaster, the Cocoanut Grove Nightclub Fire in 1942, marked the next development in burn resuscitation. Hospitals treating the victims of the fire used large volumes of plasma and crystalloid in their resuscitations. This approach was closely studied by the National Research Council in preparation for increased U.S. involvement in World War II. The combined colloid and crystalloid strategy would be battle-tested and ultimately codified in the Evans and Brooke formulas. The role for EResus in this strategy was thought optimal for people with smaller burns (e.g., <20% TBSA) and ideally after initial IV fluid resuscitation [59].

EResus was revived in the years immediately after World War II in response to the looming threat of nuclear warfare. Fearing overwhelming civilian casualties like those seen

in Hiroshima and Nagasaki, the medical community was looking for an easy-to-administer, low-resource, safe, and effective treatment for burn shock, particularly for casualties with injuries that could be managed during a catastrophic event. In 1950, the US National Institute of Health (NIH) Surgery Study Section recommended the use of oral electrolyte solution for treatment in a mass-casualty burn scenario after experiments in mouse models demonstrated equivalent results with both enteral and IV resuscitation for burns [60]. Subsequent clinical studies refined the approach to oral resuscitation by using buffered solutions to improve palatability and decrease nausea and vomiting [61,62].

After the NIH statement, several clinical studies examining EResus ensued. Most continued using buffered electrolyte solutions [63–68]. Exceptions included work by Sørenson et al., who utilized a combination of a patient-selected clear fluid with salt tablets (7.5 g tablet per liter of fluid) as well as Franke and Kock-Marburn, who utilized an oral electrolyte solution containing glucose well before the development of ORS [69,70]. The limitations of many studies published before 1980 included the use of fixed resuscitation strategies that did not scale with the size of the burn injury and solutions not optimized for maximal gut absorption. This resulted in under-resuscitation and poor outcomes for patients with very large burns [71].

There have been relatively few studies examining the EResus of people with burn injuries since the dissemination of ORS worldwide in the 1970s and 1980s. Of the studies performed after 1970, three used glucose-containing solutions for resuscitation [72–74]. Ahnefeld demonstrated that one of the major limitations of EResus was poor tolerance in patients presenting after two hours and those in clinical shock (e.g., hypotension). This could be mitigated, however, with the continuous nasogastric administration of resuscitation fluids [72]. El-Sonbaty's study comparing IV resuscitation using the Parkland formula and EResus using the World Health Organization ORS (WHO-ORS) was limited to only moderate burns, but it demonstrated no differences in key outcomes between the enteral and IV resuscitation strategies [73]. Moghazy combined the use of ORS with salt tablets and showed no difference in vital signs or urine output when compared to a control group receiving IV fluids alone [74].

## 7. Application of Enteral Resuscitation in the Austere Setting

As there have been no large randomized clinical trials to guide therapy, EResus indications, contraindications, and strategies vary widely. Resource availability may, therefore, dictate the strategy used. For patients with burns < 10–15% TBSA, oral intake ad libitum (i.e., to thirst) is usually the only resuscitation needed. For larger burns (>15–20% TBSA in adults and >10–15% TBSA in children), formal resuscitation with salt-containing fluids should be administered [75]. Lactated Ringer's remains the principal resuscitation fluid even in low-resource settings, and IV access should be established as soon as possible to facilitate both resuscitation and pain management, as indicated [75]. In the event of limited IV fluid availability, EResus becomes a vital adjunct that permits the optimization of scarce resources while achieving key resuscitation endpoints (e.g., target urine output, normal vital signs indicative of adequate end-organ perfusion) and potentially mitigating the risks of gastrointestinal mucosal barrier and immunological dysfunction. Certain patients, particularly those with smaller burns, may be able to receive all their resuscitation enterally, and those with larger burns may have decreased IV fluid requirements with supplemental enteral resuscitation.

## 8. Solutions

A wide variety of solutions can be used during enteral resuscitation, but WHO-ORS is most optimized for intestinal absorption. The newer reduced-osmolarity ORS is generally preferred to the older formulation (i.e., 245 mOsm/L rather than 311 mOsm/L) but also has less sodium (75 mmol/L rather than 90 mmol/L). Reduced-osmolarity ORS has proven to be more efficacious in patients with acute watery diarrhea and is associated with reduced stool output, reduced vomiting, and a reduced need for IV fluid; however, there are

concerns that it increases the risk of hyponatremia [76,77]. While there is some evidence of increases in mild biochemical hyponatremia (Sodium < 130 mmol/L), this does not appear to lead to increases in cases of symptomatic hyponatremia [77,78]. This may have implications for patients undergoing major burn resuscitation, among whom hyponatremia is common [79]. No EResus studies in burn patients have used the newer ORS formulation, but Sonbaty compared the old ORS formula against standard of care IV resuscitation using lactated Ringer's and found equivalent rates of hyponatremia [73].

ORS is the most ubiquitous EResus solution and is widely available globally, given its status in the *World Health Organization Model List of Essential Medicines*. If prepared sachets are not available, ORS can be made with commonly accessible ingredients using the recipe in Figure 1 (e.g., (1 L of water + 0.5 teaspoon of salt + 6 teaspoons of sugar) or (1 L of water + 0.25 teaspoon of salt + 0.25 teaspoon of baking soda + 6 teaspoons of sugar or honey)). Care must be taken when using homemade solutions, however, as mixing failures are reported in studies that used family members to mix ORS [80]. Other solutions such as rice water, thin soups, or sports drinks (supplemented with ¼ tsp of both salt and baking soda per quart bottle) may also be used during a crisis when ORS is not immediately available but are less effective given that the ratios of sodium, potassium, and glucose are not optimized for gastrointestinal absorption [81]. Additionally, large volumes of hyperosmolar solutions may generate diarrhea, particularly in patients with ineffective brush border enzymes, and hyponatremic fluids can lead to systemic hyponatremia when administered in large volumes. When not using ORS, it is important to supplement every liter of fluid with 7.5 g of salt per Sørenson's formula to avoid hyponatremia [73,75].

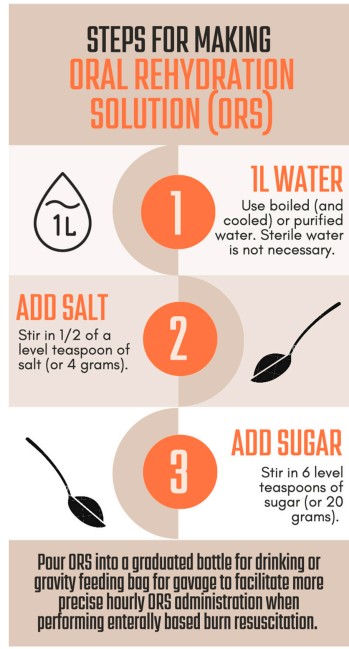

**Figure 1.** Steps for making oral rehydration solution from readily available ingredients.

Several considerations are worth mentioning, including palatability, temperature, and route of administration. Numerous factors contribute to palatability, including electrolyte composition, sweetness, flavoring, temperature, and patient preferences [82]. Palatability is negatively affected by the high sodium content of enteral resuscitation fluids (>50 mmol/L) [83]. Commercial solutions touting improved palatability (e.g., drip drop, liquid IV) should be used with caution due to their lower sodium content (40–60 mmol/L vs. 75–90 mmol/L) than WHO-ORS, which could increase the risk of developing hyponatremia. Citrus-flavored solutions have higher palatability than unflavored solutions [84,85]. Room temperature (15–20 °C) to slightly cooler fluids are generally preferred [86]. Some patients find it easier to consume through a straw, particularly those with hand and/or

face injuries. Nasogastric tube administration via either an enteral feeding pump or hourly gavage is useful for those who are intubated, have altered mental status, are unable to use their hands or arms, are without a buddy to help, or are sleeping.

## 9. Fluid Rates

EResus can be administered in accordance with fluid resuscitation formulas such as the modified Brooke (2 mL/kg/%TBSA burned, first 24 h) or Parkland (4 mL/kg/%TBSA burned, first 24 h) and adjusted according to resuscitation endpoints based on serial assessments (Figure 2). A simplified formula created by the US Army Institute of Surgical Research called the Rule of 10's can also be used to determine the initial fluid rate for adults using the following formula: first, estimate burn size to the nearest 10; second, multiply the burn size (% TBSA) by 10 for the initial rate in mL/h; third, add 100 mL/h for every 10 kg over 80 kg [87].

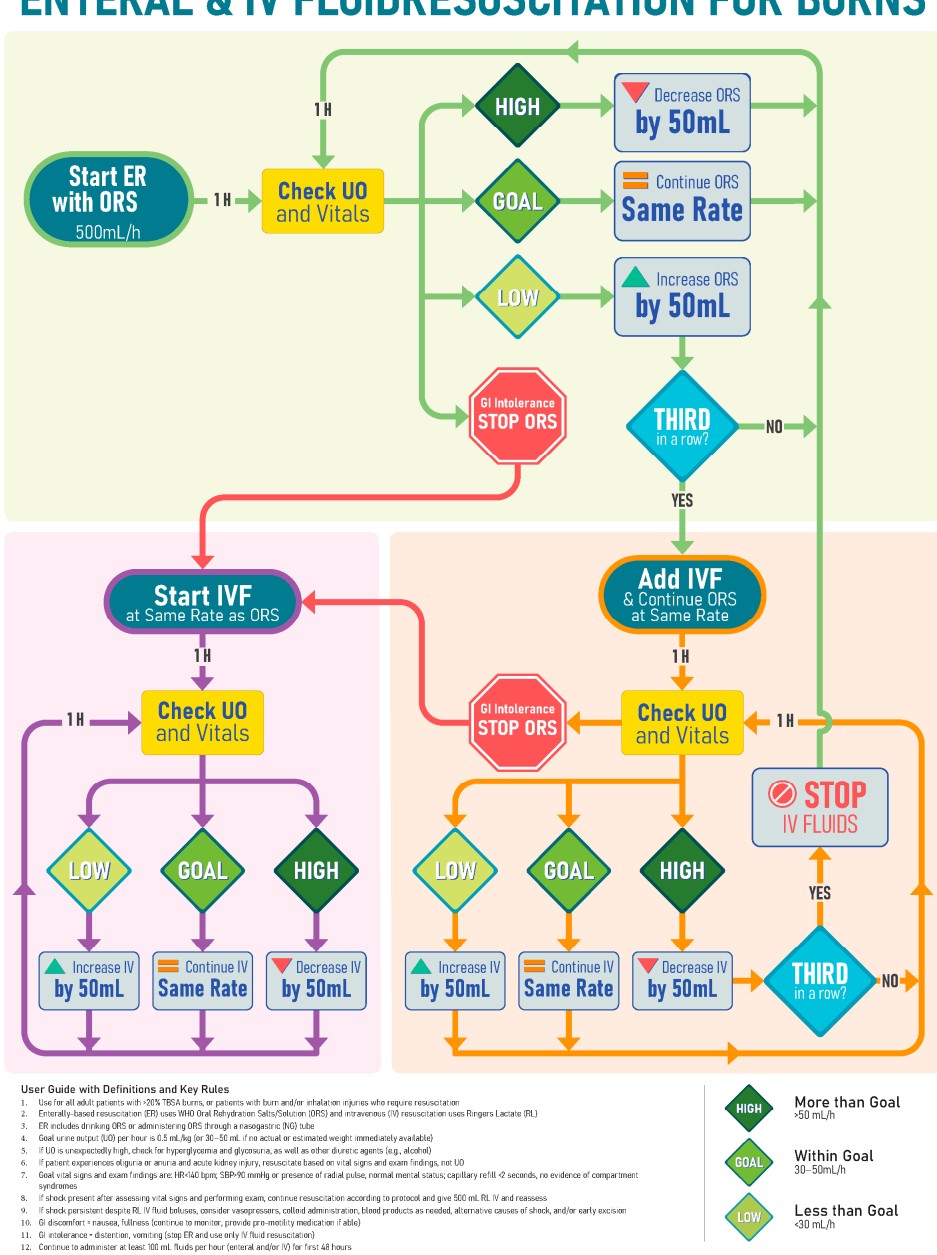

**Figure 2.** Example resuscitation strategy for use in austere settings that combines enteral and intravenous fluid and does not require estimation of burn size or calculations.

A formula based on burn size can be difficult to implement without trained burn providers and in challenging operational environments [88–90]. The International Society for Burn Injuries (ISBI) guidelines recommend that patients with major burn injuries drink 15% of their body weight (in kilograms, that is, in liters) daily for the first two days after the burn injury [75]. A WHO Technical Working Group on Burns (TWGB) proposed an alternative formula of 100 mL/kg (~10% body weight) daily [91]. As can be seen in Figure 3, the strategy proposed by the TGWB is optimized for 20–50% TBSA injuries based on the assumption that patients with ≥60% TBSA burned are unlikely to survive in a burn disaster [92]. The simplest strategy, however, is the American Burn Association prehospital strategy, which recommends basing the initial fluid rate based on patient age (500 mL/h for age > 14 years, 250 mL/h for ages 6–13 years, and 125 mL/h for age <5 years) [93]. The performance of this age-based strategy diminishes at the extremes of weight (Figure 3) and should be adjusted to one of the above formulas as more information on the patient is obtained.

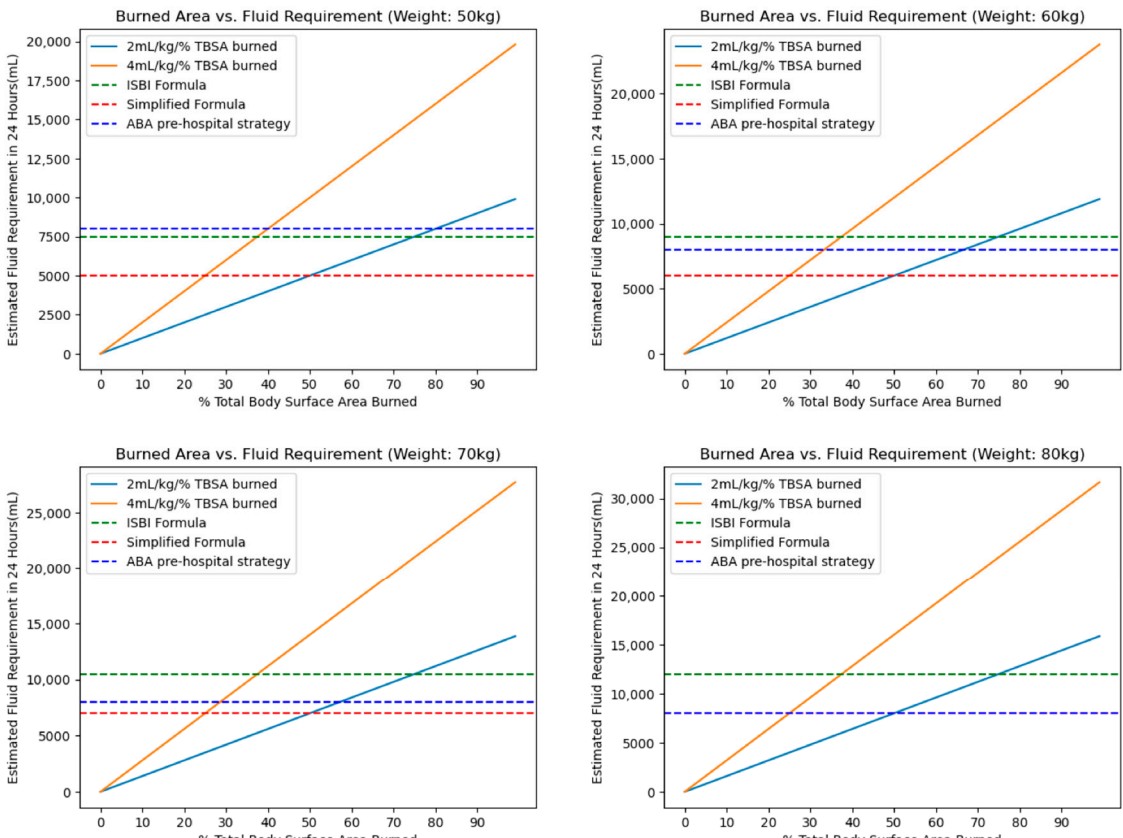

**Figure 3.** Comparison of the modified Brooke (2 mL × kg × %TBSA), Parkland (4 mL × kg × %TBSA), International Society for Burn Injuries formula, World Health Organization Technical Working Group for Burns simplified formula proposal, and the American Burn Association prehospital fluid rate for adults across a range of weights. Abbreviations: total body surface area (TBSA), International Society for Burn Injuries (ISBI), American Burn Association (ABA).

It should be emphasized that resuscitation is different from ad libitum fluid intake. Thirst is stimulated relatively late in dehydration [82]. As a result, humans are often poor at replacing ongoing fluid losses [94]. Intake should therefore be encouraged proactively and recorded using a protocol such as the one in Figure 4. If the patient develops severe gastrointestinal intolerance such as refractory nausea, vomiting, or diarrhea, then it may be necessary to stop EResus and transition to IV resuscitation. Once the GI intolerance subsides (1–2 h), enteral intake and resuscitation can typically be restarted.

| Burn Enteral Resuscitation Protocol |
|---|
| For use in burns greater than 10%–20% total body surface area (TBSA) or combined burns/inhalation injury |
| **Equipment and supplies** |
| <ul><li>Oral Rehydration Solution (ORS)</li><li>Intravenous/intraosseous access</li><li>Graduated container to measure intake</li><li>Nasogastric tube</li><li>Foley Catheter or urinal to measure urine output</li></ul> |
| **Fluid Rate** |
| Burn-size dependent<ul><li>Modified Brooke: 2mL × kg × %TBSA burn over the first 24 hours with half infused in the first 8 hours</li><li>Parkland: 4mL × kg × %TBSA burn over the first 24 hours with half infused in the first 8 hours</li><li>Rule of 10s: 10mL/h × %TBSA (to the nearest 10), add 100mL/h for every 10 kg over 80kg</li></ul>Burn-size independent<ul><li>ISBI: 150 mL/kg (15% of bodyweight) daily</li><li>Simplified: 100mL/kg daily</li></ul>Weight and Burn-size independent<ul><li>ABA prehospital strategy: 500mL/h (for >14 years old)</li></ul> |
| **End Points** |
| <ul><li>Monitor vital signs and urine output (if available) hourly</li><li>If a foley catheter is used goal urine output is 0.5mL/kg/h (or 30–50mL/h, if the patients weight is unknown)</li><li>If urine output is not available then resuscitate based on vitals signs with the goal of HR <140bpm, SBP >90mmHG or present radial pulse, normal mental status, capillary refill time <2s, and no evidence of compartment syndrome</li><li>Titrate: increase 10–20% or 50 ml/h if below goal, decrease by 10–20% or 50 ml/h if above goal</li></ul> |
| **Implementation considerations** |
| <ul><li>Obtain IV access as soon as possible</li><li>If the patient develops GI intolerance then stop ORS, administer prokinetic agent (e.g., metoclopramide), and start IV fluids at the same rate; restart in 2 hours once GI intolerance resolves.</li><li>Augment enteral resuscitation with IV fluids if shock, GI intolerance, persistently oliguria (e.g., >4 consecutive hours), and/or larger burn injury (e.g., >40% TBSA or smaller if very deep injury).</li><li>A straw may make drinking easier; if a patient cannot drink on their own (e.g., unconscious, hand burns) then a nasogastric tube should be placed. A nasogastric tube can facilitate enteral resuscitation while the patient sleeps as well.</li><li>Use a graduated container to measure intake (if one is not available one adult sip is 15mL)</li><li>Ensure adequate pain control and anti-nausea therapy throughout resuscitation</li><li>For patients arriving in shock, give a 500 mL bolus of LR in addition to enteral resuscitation; for continued shock, consider additional LR, vasopressors, colloid, and/or blood-product administration, as indicated.</li></ul> |

**Figure 4.** Example enteral resuscitation strategy. Abbreviations: American Burn Association (ABA), gastrointestinal (GI), International Society for Burn Injuries (ISBI), intravenous (IV), lactated Ringer's (LR), oral rehydration solution (ORS), total body surface area (TBSA).

## 10. Implications for Civilians in Conflict Settings

EResus has important implications for the care of civilian casualties in conflict settings. Burn-injured patients are among the most vulnerable subpopulations in war. A comprehensive analysis of humanitarian care provided by the U.S. military during the wars in Iraq and Afghanistan found that children under 12 were disproportionately affected by burn injuries (16% vs. 5% among all civilians) [95]. Children with war-related burns have been demonstrated to have mortality as high as 47%, compared to 11% mortality among all children with conflict-related injuries [96,97]. Due to the resource-intensive nature of burn care and the high associated mortality in low-resource settings, resuscitation (or comfort-focused care) of people with TBSA > 60% is one of only two criteria on which standards of trauma care vary for local nationals versus coalition forces personnel treated at deployed military treatment facilities [95]. Armed conflict in the 21st century is characterized by the increasing use of explosive weapons with wide-ranging effects on civilian populations. These munitions frequently inflict associated thermal injuries. In this context, it is critical that strategies such as EResus with the potential to improve survival among this population be pursued, scaled, and incorporated into trauma training designed for low-resource settings.

## 11. Future Directions

EResus is a practical solution for resuscitation in austere settings and a promising intervention in situations when early enteral feeding is not possible, particularly during prolonged-field-care scenarios and burn disasters. In addition to the clinical trials covered above, there are an increasing number of animal studies demonstrating some benefit with EResus [56,60,98–106]. However, limited evidence exists to create strong guidelines. Randomized controlled trials are underway in Nepal and Ghana to further our understanding of its safety, effectiveness, and implementation strategies. Additional translational research is needed as well, particularly regarding the effect of enteral resuscitation on intestinal mucosal barrier function, gut microbiome, and systemic immunoinflammatory activation.

Given the difficulty of conducting controlled trials, especially in operational settings, interventional and observational studies in other low-resource contexts will play a key role in refining therapy as it is applied on the battlefield. This will be difficult, however, and will require improvements in data collection systems. Even in large deployed medical facilities, data capture can be challenging, and this is only amplified for medical units closer to the front line. This difficulty is reflected in the diminishing quality and availability of data for U.S. soldiers treated in Role 2 (i.e., Forward Surgical Teams) and prehospital settings [107,108]. As the military focuses on prolonged field care, which encompasses EResus, data collection in austere environments will be crucial for process improvement.

Another area of ongoing research is the WHO-ORS formulation itself, which was transitioned to a low-osmolarity formula in 2002. There is now a push for the addition of resistant starches to the ORS formulation [22]. These starches would be fermented by gut bacteria into short-chain fatty acids (SCFA) that would be absorbed in the colon by active cotransport with sodium—a process similar to the way glucose is absorbed in the small intestine [109]. SCFA are also important inflammatory mediators in the gut and have been shown to improve barrier function [43]. Other groups focusing specifically on burn and trauma resuscitation have examined the addition of pyruvate to EResus formulas with promising results in animal models [102–106,110].

The interaction between early enteral feeding and enteral resuscitation will also need investigation. When gastric emptying and intestinal transit were compared between tube feeding formulas and enteral resuscitation fluids in a rat model, there was a significant delay in both gastric emptying and transit time with tube-feeding formulas [37].

## 12. Conclusions

Intravenous fluid resuscitation is a logistically difficult therapy to administer on the modern battlefield and in other austere settings. The crystalloid solutions conventionally

used in burn resuscitation are bulky and will become increasingly scarce as blood products become the primary resuscitative fluid for hemorrhage. This means that those caring for conflict casualties may need to rely on alternative strategies for resuscitation after burn injuries, such as EResus. Previous studies on animal models suggest that enteral resuscitation using salt-containing solutions is effective for up to 40% TBSA burns. Relatively few studies have been conducted with ORS, which maximizes fluid and electrolyte absorption. While further research is needed to generate the evidence base for refinement of this technique, medical personnel should provide EResus using available resources when IV fluid therapy is not available.

**Author Contributions:** Conceptualization: I.F.J., K.N. (Kiran Nakarmi), H.B.W., K.N. (Kwesi Nsaful), K.M., R.S., D.R. and B.T.S. Writing-original draft preparation: I.F.J., H.B.W. Writing-review and editing: I.F.J., K.N. (Kiran Nakarmi), H.B.W., K.N. (Kwesi Nsaful), K.M., R.S., D.R. and B.T.S. All authors have read and agreed to the published version of the manuscript.

**Funding:** This research received no external funding.

**Institutional Review Board Statement:** Not applicable.

**Informed Consent Statement:** Not applicable.

**Data Availability Statement:** No new data were created or analyzed in this study. Data sharing is not applicable to this article.

**Conflicts of Interest:** The authors declare no conflicts of interest. The views expressed in this article reflect the opinions of the authors and do not reflect the official policy or position of the U.S. Army, Department of Defense, or U.S. Government.

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
