# Peer review of "Enteral Resuscitation: A Field-Expedient Treatment Strategy for Burn Shock during Wartime and in Other Austere Settings"

_2673-1991, doi:10.3390/ebj5010003_

Round 1
Reviewer 1 Report
Comments and Suggestions for Authors
In this narrative review, the authors discuss the feasibility and appropriateness of enteral resuscitation (EResus) to treat burn shock at war or in austere settings. They draw attention to this old yet neither sufficiently studied nor sufficiently known treatment strategy.
On the whole, their manuscript is well written and easy to read. I think it would benefit from the following adjustments.
- The authors refer to cholera because this disease, among other causes of infectious diarrhea, prompted the development of ORS. But can cholera-related acute dehydration and burn shock be relevantly compared? The differences between the content of fluids lost with watery diarrhea on the one hand and plasma lost by capillary leakage and exsudation in severe burns on the other should be addressed, as well as whether replacing lost fluids by ORS can soundly apply to both situations, considering ORS composition and available literature about EResus.
Most importantly, minimizing the risk of hyponatremia is not sufficiently addressed in the manuscript, as compared with longer developments of more limited clinical relevance.
- In section 3-LOGISTICAL CONSTRAINTS, is "locally sourced clean (not sterile) water" dependable in austere situations? In an actual war or disaster, access to water, and especially to clean water, can be disrupted. It can require the deployment of mobile water treatment units. This could limit the usability or the timeliness of EResus.
- In section 5-GUT PHYSIOLOGY IN BURNS, the developments about enteral nutrition (l.165-178) are partly off-topic and could be shortened.
- In section 10-FLUID RATES, was the strategy summarized on fig.2 designed for this paper? Most importantly, the figure looks simple but its applicability is questionable. Would hourly adjusting flow rates of either enteral or even intravenous fluids by plus or minus 50 mL/h be feasible in settings where the availability of flow rate control devices is unlikely?
On the whole, I agree with the authors that EResus with ORS makes sense at war or in austere settings and that it is worthy of further research. But I suggest their proposed EResus strategy be refined to address the lack of flow rate measurement devices in the target settings.
- Minor details:
l.42 - Please rewrite for clarity
l.152-164 - Should be more concise
l.280 - Consider a more appropriate section title
l.331-333 - Please rewrite for clarity
Author Response
You have made several insightful comments.
Point 1) Enteral resuscitation with ORS was selected because it has been successful in the treatment of another disease characterized by massive loss of intravascular volume. The contents of burn fluid losses and cholera are not exact. In regards to sodium, however, they are comparable with burn blister fluid containing ~130-160mmol/L (Evaluation of burn blister fluid. Heggers et al.) and cholera losses containing ~130mmol/L (Dysnatremia in Gastrointestinal Disorders. Do et al.). The concern about hyponatremia with reduced osmolarity ORS is primarily in cholera patients due to their increased sodium losses when compared to other causes of acute watery diarrhea. In our revision we have included additional safety studies demonstrating a low rate of symptomatic hyponatremia in cholera patients resuscitated with the new ORS formulation and have highlighted the results of the El-Sonbaty study demonstrating no difference in hyponatremia between ORS resuscitation and standard of care resuscitation with lactated ringers in moderate burns.
Point 2) We agree that providing clean drinking water is difficult in a chaotic wartime or disaster scenario and that at a certain level of disruption even enteral resuscitation would become impossible. There are a multitude of methods to provide clean drinking water and a medical facility that can sustain its staff would likely have some access to clean water, which makes enteral resuscitation attractive. This is a much easier logistic hurdle than the alternative, IV fluids, which can suffer from shortages even in well-resourced settings (Facing the Shortage of IV Fluids — A Hospital-Based Oral Rehydration Strategy Patiño et al.).
Point 3) We have included this section because the altered integrity and function of the GI tract are an important consequence of burn injury that may impact the feasibility of enteral resuscitation as a strategy. While the exact effects of ORS on the GI tract of burn injured patient has not been studied, we felt evidence that these negative effects can be mitigated by oral intake is important to consider when evaluating enteral resuscitation as a strategy.
Point 4) Titration of enteral resuscitation has not been previously study. The diagram in figure 2 was produced by some of the authors for the use in an ongoing study that will provide insight into whether this is a viable strategy. Paced fluid intake has been successful in other protocols (Facing the Shortage of IV Fluids — A Hospital-Based Oral Rehydration Strategy Patiño et al.), however, using either measuring cups or estimation of intake (2 large sips = 30mL) and a timer.
Minor details) These changes have been made.
Reviewer 2 Report
Comments and Suggestions for Authors
In the manuscript “Enteral resuscitation: a field-expedient treatment strategy for burn shock during wartime and in other austere settings” investigators review protocols, advantages and disadvantages of enteral resuscitation. The context of wartime burns and resource constraints is great and a perfect fit for this special issue. Largely, the authors present logical and well written content and should be congratulates. Some things to consider are given here:
- -The authors should mention the importance of checking gastric residuals and the need for nurse or medic technical hands to do so. Other studies have shown that vomiting is fairly rare, so the most common GI intolerance sign may be high residuals.
- - There is a growing body of knowledge from animal models about the effects of enteral resuscitation on burn pathophysiology, including effects on organs other than the gut. That warrants inclusion here.
- - Some more conversation about the gut microbiome in burns might increase access of this manuscript given the novelty of this subject.
- - The “Solutions” and “Logistical considerations” sections could be merged, and should expand on palatability, to discuss ORS compared to other commercially available solutions (e.g., Drip Drop, Liquid IV).
- - Figure 3 is fairly blurry in its current form.
Author Response
We appreciate your perceptive comments.
Point 1) Many patients receiving enteral resuscitation will not have a nasogastric tube and there would be no easy way to check gastric residuals. Gastric residual monitoring is also associated with an increase in treatment interruptions (Is monitoring of gastric residual volume for critically ill patients with enteral nutrition necessary? A meta-analysis and systematic review Feng et al.), which could be dangerous during burn resuscitation, so we have not included it in our recommended strategy.
Point 2) We have focused primarily on human clinical trials of enteral resuscitation to provide our readers with the most clinically relevant picture. We would like to acknowledge the many articles reporting the benefits of enteral resuscitation in animal models as well and have done so in the future directions section of our revised manuscript.
Point 3) Gut microbiome changes in response to burn injury is an interesting field that is relatively understudied. We chose to limit our discussion on this topic due to the lack of both human studies and studies examining the interaction of enteral resuscitation on the microbiome.
Poin 4) These sections have been combined. We have also discussed more palatable options such as drip drop and liquid IV to note that one of the ways that they achieve increased palatability is by decreasing the sodium content, which may increase the risk of hyponatremia.
Point 5) I believe that the blurry appearance of figure 3 is due to how Microsoft Word scales the font in the image. The legibility is significantly better in the pdf version.
Reviewer 3 Report
Comments and Suggestions for Authors
MS. Review (ebj-2623562-peer-review) Enteral resuscitation: a field-expedient treatment strategy for burn shock during wartime and in other austere settings by Jones and colleagues.
This is a narrative review on the topic of enteral fluid resuscitation of burn shock during wartime and in austere settings.
This is a very well thought of review on an important topic of burn care. This issue has previously been poorly addressed and the value of the present review is significant. The manuscript is well organized, clearly written. The text is supported by good Figures and illustrations.
There are some minor issues that needs to be addressed.
First, the figures and illustrations, included do not seem to be all drawn by the authors. It is important to ascertain that proper attention to copywrite is maintained. Please provide information on this and possibly support if needed that the copywrite issue is settled.
Second, a rather recent publication that suggest added value of enteral fluid resuscitation to that beyond intravenous resuscitation has been published which may be commented upon (Enteral resuscitation with oral rehydration solution to reduce acute kidney injury in burn victims: Evidence from a porcine model by Belinda I. Gómez, PLOS ONE, May 2018.)
Author Response
We appreciate your comments.
Point 1) All figures used in this article were produced by the authors and have not been used in any previously published work.
Point 2) While we have focused mainly on human clinical trials in this paper to provide our readers with the most clinically relevant data, we do want to acknowledge the animal studies, like the one suggested, that also support the use of enteral resuscitation. This has been done so in the future directions section of the revised manuscript.